# Efficient Synthesis and In Vitro Hypoglycemic Activity of Rare Apigenin Glycosylation Derivatives

**DOI:** 10.3390/molecules28020533

**Published:** 2023-01-05

**Authors:** Lin Zhao, Yuqiong Pei, Guoxin Zhang, Jiayao Li, Yujie Zhu, Mingjun Xia, Ke Yan, Wen Mu, Jing Han, Sen Zhang, Jinao Duan

**Affiliations:** Jiangsu Collaborative Innovation Center of Chinese Medicinal Resources Industrialization, Jiangsu Key Laboratory for High Technology Research of TCM Formulae, Crich, 138 Xianlin Road, Nanjing 210023, China

**Keywords:** apigenin, glycosylation, succinylation, *Bacillus licheniformis*, hypoglycemic

## Abstract

Apigenin is a natural flavonoid with significant biological activity, but poor solubility in water and low bioavailability limits its use in the food and pharmaceutical industries. In this paper, apigenin-7-*O*-β-(6″-*O*)-d-glucoside (AG) and apigenin-7-*O*-β-(6″-*O*-succinyl)-d-glucoside (SAG), rare apigenin glycosyl and succinyl derivatives formed by the organic solvent-tolerant bacteria *Bacillus licheniformis* WNJ02 were used in a 10.0% DMSO (v/v) system. The water solubility of SAG was 174 times that of apigenin, which solved the application problem. In the biotransformation reaction, the conversion rate of apigenin (1.0 g/L) was 100% at 24 h, and the yield of SAG was 94.2%. Molecular docking showed that the hypoglycemic activity of apigenin, apigenin-7-glucosides (AG), and SAG was mediated by binding with amino acids of α-glucosidase. The molecular docking results were verified by an in vitro anti-α-glucosidase assay and glucose consumption assay of active compounds. SAG had significant anti-α-glucosidase activity, with an IC_50_ of 0.485 mM and enhanced glucose consumption in HepG2 cells, which make it an excellent α-glucosidase inhibitor.

## 1. Introduction

Apigenin (4′, 5, 7-trihydroxy flavone) is a flavonoid that is present in many leafy foods such as parsley, orange, tea, chamomile, and others [1]. It has recently been found to have significant anti-inflammatory, anti-oxidant, antitumor, and hypoglycemic activity [2]. The 4′, 5, 7 trihydroxy and C2, C3 double bonds are responsible for apigenin’s pharmacological activity, low toxicity, and stability, which are desirable characteristics for drug development [3]. However, its low solubility in water reduces its bioavailability and limits clinical application. Glycosylation improves water solubility and bioavailability [4], and glycosylated flavonoids that are found in many fruits and vegetables have anticancer activity, and activity against cardiovascular, and neurodegenerative diseases. Antimalarial phytochemicals, including apigenin-7-O-glucoside, luteolin-7-O-glucoside, and others have predicted high-binding energy for the 3C-like protease of COVID-19 [5], and are considered useful for developing candidate antiCOVID-19 drugs. Unlike apigenin, which is easily extracted from plant stems and leaves, nonedible parts of plants, and non-medicinal parts of herbal prescriptions [6], apigenin-7-O-glucoside (AG) extraction is difficult, inefficient, and expensive. Developing a new method of synthesizing apigenin glycoside derivatives is of great research interest.

Glycosylation reactions can be carried out by chemical and biological means [7]. Compared with synthetic glycosylation, biotransformation catalyzed by glycosyltransferases or produced by microorganisms avoids repetitive, multistep, complex, and dangerous steps. The products of biotransformation are also more easily isolated and purified than products of chemical synthesis [8,9]. The 7-hydroxycoumarin derivatives esculetin, skimmin, and herniarin produced by *Bacillus licheniformis* DSM 13 and UDP-glycosyltransferase YjiC were found to have greater antibacterial and anticancer activity than 7-hydroxycoumarin [10]. However, bacteria and enzymes are easily inactivated by organic solvents during biotransformation reactions, which has led to searches for extremophiles [11,12]. Our research group has screened extremophiles and developed a biosynthetic method of glycosylating and succinylating flavonoids by organic solvent-resistant bacteria. The method improves the bioavailability of flavonoids and facilitates the development of flavonoids with potential clinical application [4,13]. Succinylation has also been found to enhance the water solvency and bioavailability of compounds and can be accomplished by a strain of *Bacillus amyloliquefaciens* [13]. With enzyme engineering, synthesizing succinyl groups also suffer from cumbersome experimental procedures and specificity of donor substrates, which highlights the advantages of exclusive strain biotransformation [14].

Chronic hyperglycemia above a certain threshold triggers diabetes, which is a systemic disorder of the endocrine system that has become a substantial threat to human health. The low cost and high therapeutic effectiveness of glucosidase inhibitors have led to clinical trials of novel compounds that have been the subject of a recent review [15]. Recent evidence indicates that polysaccharides have a key role in preventing diabetes and its complications [16], that glycosides have good hypoglycemic activity, and that natural compounds have broad prospects for clinical development. Recent evidence supports the effectiveness of apigenin-7-O-glucoside and dietary flavonoid glycosides as α-glucosidase inhibitors and insulin sensitizers. SAG is a novel compound synthesized by our team for the first time, but the hypoglycemic activity of SAG has not been reported [17].

In this study, we developed a synthesis method using *B. licheniformis* WNJ02 to efficiently biotransform apigenin into AG and SAG (as shown in Figure 1), which are rare apigenin glycosylation derivatives that overcame limitations of the use of apigenin. Additionally, assays of glucose consumption in HepG2 cells and an in vitro anti-α-glucosidase activity were developed. Molecular docking was used to explain the structure–function relationships of apigenin and its derivatives with α-glucosidase. The aim was to support development of novel clinical drugs and lay the foundation for the development of effective antidiabetic drugs.

## 2. Results

### 2.1. Biotransformation and Characterization of Apigenin

*B. amyloliquefaciens* FJ18 [4,18] and *B. licheniformis* ZSP01 [13] are organic solvent-resistant strains that produce glycosyltransferase and can biotransform apigenin. Production of SAG by *B. amyloliquefaciens* FJ18 in a nonaqueous phase has been previously described [18]. *B. licheniformis* WNJ02 (CCTCC NO:M2018394) was adaptively domesticated by *B. amyloliquefaciens* FJ18. WNJ02 was glycosylated with the 7-phenolic hydroxyl group of apigenin to form apigenin-7-*O*-β-(6″-*O*)-d-glucoside, and the 6-hydroxyl group of the glucose group in AG was succinylated to form SAG. The conversion rate of apigenin to SAG by WNJ02 was 94.2%. The reaction schema is shown in Figure 1a. High-resolution ESI-MS of the SAG product (Appendix A) detected a molecular ion [M+H] ^+^ with a mass of 533.13, with apigenin at m/z 271.06, presumably linking a glucose and a succinyl group at different sites to obtain m/z 533.13 and m/z 534.13, indicating an empirical formula of C_25_H_24_O_13_. The ^1^H NMR and ^13^C NMR results are shown in Table 1, Figure 1b, Appendix A, which show the characteristics of a glucose group and a succinyl group. HMBC (Appendix A) confirmed that the hydroxy group at O-6″ of the glycosyl group was succinylated to form apigenin-7-O-β-(6″-O-succinyl)-d-glucoside, namely SAG. Our group successfully developed the first biological method to form succinylated derivatives.

### 2.2. Molecular Docking

Acarbose has been used as a first-line antidiabetic drug since 1990, but screening of potential natural α-glucosidase inhibitors is ongoing [19]. Current studies of natural α-glucosidase inhibitors are mostly in vitro and do not meet the needs of clinical new drug development. Acarbose treats postprandial hypoglycemia by reducing the digestion of complex carbohydrates and inhibiting human and bacterial glucosidases, but acarbose resistance limits clinical use [20]. Apigenin and AG have good hypoglycemic activity, but whether rare glycosylation derivatives have optimal activity needs further study. We screened the rare, potential α-glucosidase inhibitors by molecular docking. Preliminary molecular docking results (Table 2) found that apigenin, AG, and SAG had better binding activity to α-glucosidase (−9 kcal/mol, −8.9 kcal/mol, −9.4 kcal/mol, respectively), than acarbose (−8.0 kcal/mol). Apigenin formed hydrogen bonds with residues ARG430, THR141, ARG513, PRO466, and ILE517 of 7d9b, electrostatic interaction with residues ARG430 of 7d9b, and hydrophobic interactions with THR141, ALA518 (Figure 2a). AG interacts with 7D9B, mainly through the formation of hydrogen bonds as well as hydrophobic forces, formed hydrogen bonds with residues GLU349, ASP440, and ARG488, and hydrophobic interactions with VAL346, TRP376, HIS439, and HIS214. (Figure 2b). SAG formed hydrogen bonds with residues ARG343, ASP440, HIS256, ASP484, and GLU296 of 7d9b, electrostatic interactions with residues GLU374 and ASP440 of 7d9b, and hydrophobic interactions with PHE306, VAL346, and TRP376 (Figure 2c). Acarbose formed hydrogen bonds with 7d9b residues ASP142, GLY245, HIS242, LYS366, ASP338, ASP340, ARG247, GLN195, and ARG244 (Figure 2d). Hydrogen substantially increased the stability of active compounds with α-glucosidase, and tight binding of ligands and enzymes is enhanced by hydrophobic and electrostatic bonds [21]. Apigenin was reported to have better hypoglycemic activity than apigenin-7-*O*-β-(6″-*O*)-d-glucoside [22], which is in line with the molecular docking results. The results show that SAG had the best binding activity, suggestive of the best hypoglycemic effectiveness. Consistent with the activity results, it is speculated that the succinyl group improves hypoglycemic effectiveness, which suggests that succinylation may affect hypoglycemia [23]. To verify the results of molecular docking, the in vitro activity assays were conducted.

### 2.3. Water Solubility and Anti-α-Glucosidase Assay of Apigenin and Derivatives

The water solubility of apigenin is 0.005 g/L and that of SAG is 0.968 g/L, which is over 174 times greater. Glycosidation and succinylation significantly improved the poor water solubility of flavonoids. The hypoglycemic activity of acarbose and different concentrations of apigenin, AG, and SAG were verified experimentally. As shown in Table 3, acarbose, a clinically approved α-glucosidase inhibitor, had an IC_50_ of 0.554 mM. The IC_50_ of apigenin succinyl glycoside was 0.485 mM, which is lower than that of apigenin (0.525 mM) and AG (0.532 mM). α-glucosidase inhibition was concentration-dependent and reached 88.89% at a SAG concentration of 1 mM. As the anti-α-glucosidase assay results verified the docking results, preliminary experiments were conducted in cell cultures to determine whether SAG has drug development prospects.

### 2.4. Cell Viability Assay of Apigenin and Derivatives


The viability of HepG2 cells was detected by the CCK-8. In a 96-well plate, HepG2 cells were cultured with normal serum-containing high glucose DMEM medium for 48 h, and the cells grew to about 70%. After 24 h of administration, the cells in the blank group grew to about 90%. The proliferation activity of HepG2 cells was detected by the CCK-8. The cell viability of different concentrations of apigenin succinyl glycoside was 95.55%, 98.87%, and 99.19%, respectively (Figure 3). Compared with apigenin and AG, SAG has almost no cytotoxicity. Therefore, the three concentrations of 5, 10, and 20 μM can be used in glucose consumption experiments, and 10 μM apigenin and AG are selected for the control group in glucose consumption experiments.

### 2.5. Glucose Consumption Assay of Apigenin and Derivatives

Metformin reduces fasting blood sugar and postprandial blood sugar, while acarbose acts to reduce postprandial blood sugar. We compared the hypoglycemic activity of metformin and rare apigenin glycoside compounds at the cellular level because it has been reported that the hypoglycemic activity of apigenin extracts and derivatives is similar to or superior to metformin [23,24]. Glucose concentration was assayed in the control and experimental groups by the glucose oxidase method, and the effects of the tested drugs on the glucose consumption of HepG2 cells was calculated and is shown in Figure 4. The cells consumed glucose in the medium during 24 h starvation, and drug intervention changed the amount of glucose consumed. A high glucose content of the medium indicates low glucose consumption by the cells. during starvation. Drug intervention promoted glucose consumption indicative of hyperglycemic activity. SAG at a concentration of 10 μM resulted in a significantly lower glucose concentration (0.2089 mM) than apigenin (0.2685 mM) or AG (0.3655 mM) at the same concentration. The glucose consumption rate of 57.09% achieved with SAG was significantly higher than that achieved by metformin hydrochloride (36.35%) at 1000 μM (Figure 4 and Figure 5). The glucose consumption observed with apigenin succinyl glycoside was concentration-dependent, reaching 66.09% at 20 μM. The results fully reflect the novel drug development prospects of apigenin and its derivatives, especially SAG.

## 3. Discussion

In a previous laboratory study, we isolated functional strains with glycosylation and succinylation activity. In this study, we evaluated the biotransformation activity of *B. licheniformis* WNJ02, which was derived from *B. amyloliquefaciens* FJ18 [4]. Microbial biotransformation is simpler and more efficient than chemical synthesis [25,26]. Bacteria and fungi have highly specific enzyme systems that can produce specific structural modifications such as glycosylation, succination, isopentenylation, and other modifications [27]. Highly efficient enzymatic biotransformation can be carried out in nonaqueous systems by organic solvent-resistant bacterial strains. In this study, AG and SAG, two rare glycosylation derivatives, were efficiently produced from apigenin by *B. licheniformis* WNJ02. Glycosylation resulted in the formation of AG and SAG which are a hundred times more soluble in water than apigenin, which is also expected to increase bioavailability.

Diabetes is now one of the most serious diseases worldwide, and as it is not curable, developing drugs that effectively treat diabetes is a research hotspot. The most effective way to treat diabetes is to inhibit α-glucosidase activity and the activity of other digestive enzymes that delay glucose absorption [28]. Plant extracts, especially flavonoids that inhibit α-amylase and α-glucosidase are thought to be the active components in natural plant extracts that are used to treat diabetes [29,30,31]. Apigenin is a flavonoid known to have biological activity. It has been reported that apigenin and AG inhibit α-glucosidase and have hypoglycemic activity [32,33]. We compared the activities of apigenin and its derivatives in inhibiting α-glucosidase in in vitro experiments, which confirmed that the glycosylation derivatives AG and SAG had significant hypoglycemic activity. Compared with acarbose and metformin, which are currently in clinical use, apigenin and SAG had better hypoglycemic activity than AG for fasting or postprandial use. The low toxicity and hypoglycemic activity of SAG favor new drug development. However, the hypoglycemic mechanism of SAG needs further study.

Apigenin and AG have been evaluated as phytochemical agents with activity against SARS-CoV [34] and may have potential applications in addition to diabetes. Apigenin aglycone is also found in plant stems and leaves, and this study supports the development of nonedible parts of plants rich in such ingredients and in non-medicinal parts of traditional Chinese medicine resources. We plan to establish an exclusive molecular library of succinyl compounds and to screen for more effective active ingredients with new technologies to develop effective drugs with hypoglycemic and clinical activity.

This study describes a method to obtain apigenin glycoside derivatives with apigenin aglycone as a substrate. Glycosylation increases the number of hydroxyl groups of precursor compounds, significantly improves the activity of compounds, increases the commercial value of precursor compounds, and reduces the cytotoxicity of biosynthesized compounds. Production by *Bacillus licheniformis* in a non-aqueous phase was developed by our team, and the yield of succinyl apigenin was 95% in 24 h. Compared with the existing systems, this reaction is greener and much more efficient. Given the good hypoglycemic activity of the substrate apigenin and its succinyl apigenin derivatives, the study measured the α-glucosidase activity and compared the cytotoxicity of the three derivatives in HepG2 test cells. SAG had the best biocompatibility in terms of hypoglycemic activity, compared with apigenin. Efficient, green biological production of succinyl apigenin and its hypoglycemic activity will have broad market prospects.

## 4. Materials and Methods

### 4.1. Materials

*B. licheniformis* WNJ02 was provided by Jiangsu Collaborative Innovation Center for Industrialization of Traditional Chinese Medicine Resources. The flavone standards, including apigenin and apigenin-7-O-glycoside, were purchased from Jiangsu Zelang Meiditech Co. (Nanjing, China), and dissolved in dimethyl sulfoxide 12 g/L (DMSO; Beyotime, Nanjing, China) as a stock solution for the experiments. Peptone, yeast extract, NaCl, Na_2_HPO_4_, KH_2_PO_4_, and sucrose in microbiological media were from Sinopharm Group Chemical Reagent Co., (Beijing, China). High-performance liquid chromatography (HPLC) grade solvents were from Sigma (San Francisco, CA, USA), and methanol and formic acid in analytical grade were from Sinopharm Group Chemical Reagent Co., (Beijing, China).

p-nitrophenyl-α-d-glucopyranoside (p-NPG) and α-glucosidase were purchased from Macklin (Shanghai, China). HepG2 was purchased from Shanghai Cell Research Institute (Shanghai, China). Dulbecco’s modified Eagle medium (DMEM), fetal bovine serum (FBS), penicillin, and streptomycin were purchased from Gibco (Invitrogen, Carlsbad, CA, USA). The assay of cell viability was estimated with a Cell Counting Kit-8 (APEXBIO, Houston, TX, USA). The glucose consumption was estimated using the glucose oxidase method with a glucose assay kit (Nanjing Jiancheng Bioengineering Institute, Xuanwu District, Nanjing, China).

An Infinite F50 microplate reader (Tecan, Männedorf, Switzerland), and Waters 2695 high-performance liquid chromatograph (Waters 2996 PDA monitor; Waters, Milford, MA, USA) were used.

### 4.2. Biotransformation of Apigenin in Aqueous Hydrophilic Media

*B. licheniformis* WNJ02 was grown on Luria Bertani medium consisting of 10 g/L peptone, 5 g/L yeast extract, and 10 g/L NaCl at pH 7.0 for 24 h, at 30 °C and 180 r min^−1^. The reaction was carried out in a 150 mM Na_2_HPO_4_/NaH_2_PO_4_ buffer at pH 8.0 with 20 g/L sucrose, 1.0 g/L apigenin, and 10% DMSO. The concentration of WNJ02 was measured by the optical density at 600 nm (OD_600_).

### 4.3. Quantitative Analysis and Characterization of Compounds

AG and SAG and fermentation bacteria were separated on a Macro Porous Resin AB-8 column. Two dried products (purity > 99.0%) were produced from the eluted fraction following evaporation. The concentration of the compounds was determined with a Waters 2695 Alliance HPLC system (Waters, USA) equipped with an ACQUITY UPLC BEH C_18_ column (1.7 μm 2.1 × 100 mm). Elution was carried out at a flow rate of 0.4 mL·min^−1^ with 0.1% formic acid water (A)/acetonitrile (B) in the following gradient mode: 0–2 min, 95%–95% B; 2–12 min, 95%–5% B; 12–13 min, 5%–5% B; 13–13.5 min, 5%–95% B; 13.5–16 min, 95%–95% B. UPLC/ESI-MS was carried out on a Waters UPLC-TOF mass spectrometer with an ACQUITY UPLC BEH C_18_ column (1.7 μm 2.1 × 100 mm). Pure apigenin conversion products dissolved in DMSO-d6 were analyzed by ^1^H NMR and ^13^C NMR spectra with a Bruker AV-400 spectrometer (Bremen, Germany) at 400 MHZ. Additionally, heteronuclear multiple bond correlation (HMBC), heteronuclear multiple quantum coherence, and distortionless enhancement by polarization transfer were also performed.

### 4.4. Water Solubility

An apigenin conversion product solution with a concentration of 0.2 g/L was prepared by placing 10 mg of product in a 50 mL volumetric flask and adding methanol to a constant volume of 50 mL. Portions of the solution were diluted to prepare standards with concentrations of 0.02 g/L, 0.04 g/L, 0.08 g/L, 0.12 g/L, and 0.16 g/L. The standard solutions were assayed by HPLC, and the peak area of each concentration sample was used to make a standard concentration curve. The solubilities of apigenin and SAG in saturated aqueous solution were determined at 37 °C.

### 4.5. Molecular Docking

To better understand the molecular interactions between ligands and receptors, docking experiments were performed using compounds (apigenin and derivatives) with α-glucosidase by Autodock 4.2. The crystal structure of α-glucosidase was obtained from the Protein Data Bank (PDB ID: 7D9B). The three-dimensional (3D) structure of ligands was prepared with Chem3D 19.0, and the energy was minimized by AutoDock Tools 1.5.6. The molecular docking results were analyzed by the binding energy of active compounds with α-glucosidase. The procedures were repeated three times and the molecular docking results were visualized by Biovia Discovery Studio 2019 (https://www.3ds.com/products-services/biovia/products/molecular-modeling-simulation/biovia-discovery-studio/, accessed on 18 March 2022) [21].

### 4.6. Anti-α-Glucosidase Assay

Compounds (25 μL) were preincubated in 160 μL of 0.05 M, pH 6.8 phosphate-buffered saline (PBS) with 10 μL of 0.1 mM, 0.5 mM, or 1 mM glycosidase in 96-well plates at 37 °C for 10 min before the assay. After 15 min incubation, the wells were rinsed with 5 μL 4-Nitrophenyl-β-d-glucopyranoside (pNPG) in PBS 0.05 M, pH 6.8, 12.5 mM, and the absorbance was measured at 405 nm. Percent inhibition was calculated as:(1)$$\mathrm{Inhibition}\;(\%)=[1-\frac{\mathrm{As}-\mathrm{Asb}}{\mathrm{Ac}-\mathrm{Acb}}]\;\times \;100$$
where As, Asb, Ac, and Acb are the absorbance of the sample, sample blank, control, and control blank. Acarbose was the positive control. The assay procedure was repeated three times [35].

### 4.7. Assay of Cell Viability

The assay of cell viability was conducted according to the Cell Counting Kit-8 (CCK8) method [36] as previously described using human hepatoma HepG2 cell lines. HepG2 human hepatoma cells were cultured in high glucose Dulbecco’s minimum Eagle medium (DMEM) containing 15% fetal bovine serum (FBS), 100 U/mL penicillin, 100 μg/mL streptomycin, and 1 mM sodium pyruvate. HepG2 cells in logarithmic growth phase were inoculated in a 96-well plate at a concentration of 1 × 10^5^ cells/well. After culturing for 48 h at 37 °C, the medium was replaced with fresh medium containing 5, 10, or 20 μM of apigenin, AG, or SAG. The HepG2 cells were co-cultured for 24 h before adding 10 μL CCK-8 solution for 2 h. Cell viability was measured at 450 nm. The percentage of viable cells was calculated as:cell viability (%) = As/Ac × 100(2)
where As and Ac are the absorbance of the sample and controls.

### 4.8. Assay of Glucose Consumption

Glucose consumption was measured as previously described, with slight modification [21]. Briefly, HepG2 cells were cultivated in 24-well microplates at a density of 1 × 10^5^ cells/well. DMEM free of FBS was added to the cells after 24 h incubation. The supernatants were discarded after 24 h, and a solution containing 1.0 mM metformin, 10 μM apigenin, 10 μM AG, or 5, 10, or 20 μM SAG. Metformin (1.0 mM) was the positive control. Glucose consumption was measured with a test kit after 24 h incubation.

### 4.9. Statistical Analysis

Data were reported as means ± standard deviation using SPSS 22.0 (IBM Corp., Armonk, NY, USA). The statistical analysis was performed with GraphPad Prism 8.01 (GraphPad Software, San Diego, CA, USA) using Student’s *t*-test. *p*-values < 0.05 were considered statistically significant.

## Figures and Tables

**Figure 1 molecules-28-00533-f001:**
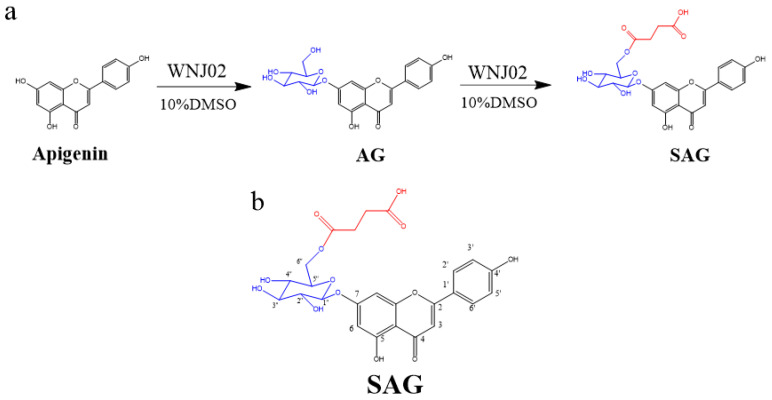
(**a**) Synthesis of apigenin-7-*O*-β-(6″-*O*)-d-glucoside (AG) and apigenin-7-*O*-β-(6″-*O*-succinyl)-d-glucoside (SAG), (**b**) The structure of SAG with numbers.

**Figure 2 molecules-28-00533-f002:**
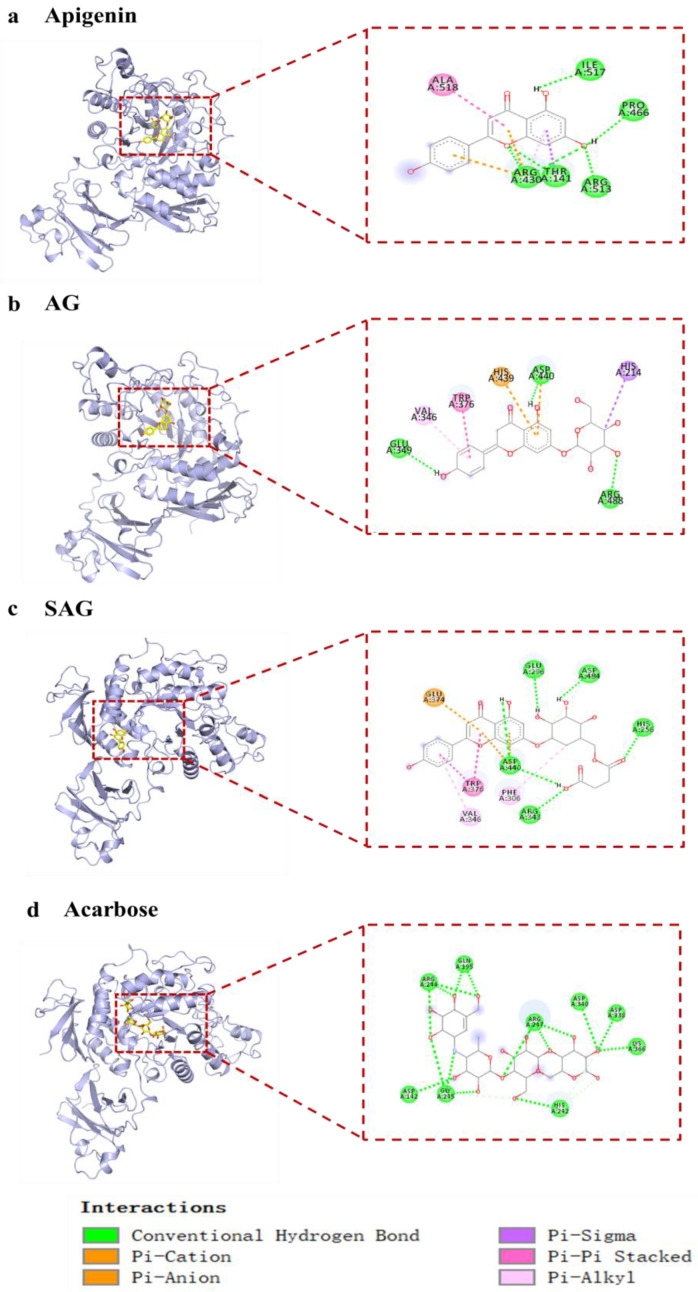
The interaction of active compounds with α-glucosidase ((**a**) apigenin, (**b**) apigenin-7-*O*-β-(6″-*O*-succinyl)-d-glucoside glucoside (AG), (**c**) apigenin-7-*O*-β-(6″-*O*-succinyl)-d-glucoside (SAG), (**d**) acarbose).

**Figure 3 molecules-28-00533-f003:**
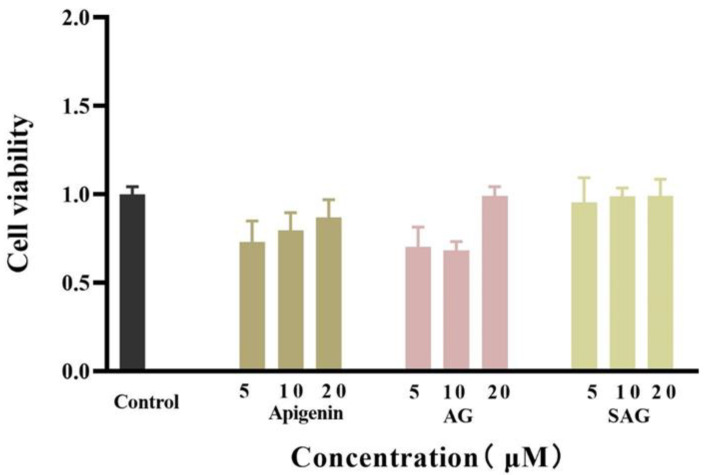
Effects of the different doses of apigenin, apigenin-7-*O*-β-(6″-*O*)-d-glucoside (AG), and apigenin-7-*O*-β-(6″-*O*-succinyl)-d-glucoside (SAG) on the cell viability of HepG2.

**Figure 4 molecules-28-00533-f004:**
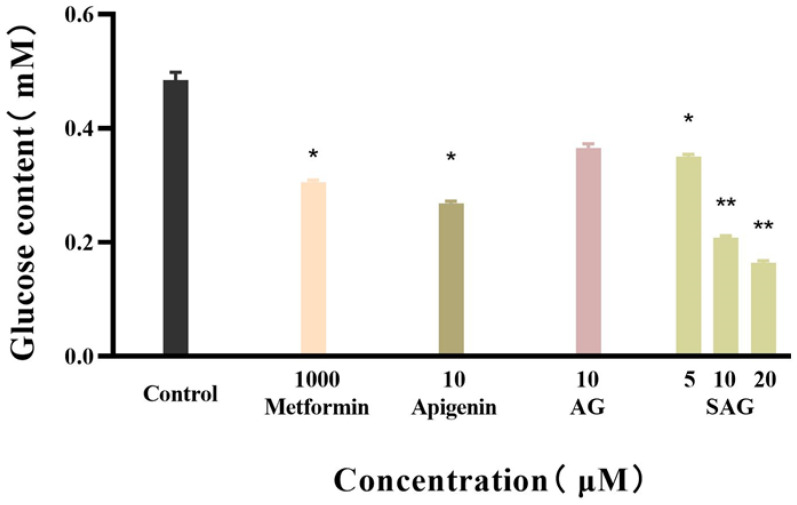
Glucose content of the apigenin and derivatives (* *p* < 0.05 vs. control group, ** *p* < 0.01 vs. control group, n = 3).

**Figure 5 molecules-28-00533-f005:**
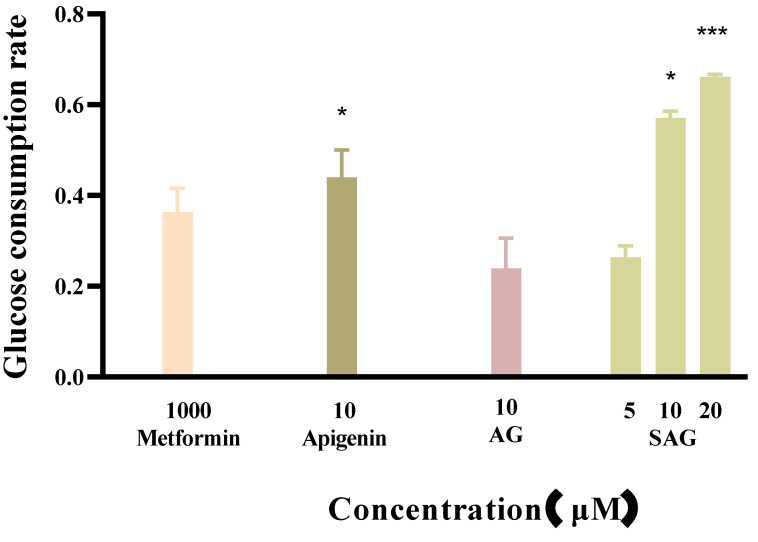
Effects of the apigenin and derivatives on glucose consumption (* *p* < 0.05 vs. metformin group, *** *p* < 0.001 vs. metformin group, n = 3).

**Table 1 molecules-28-00533-t001:** ^1^H (DMSO-d6, 400 MHz and ^13^C (DMSO-d6,400 MHz) NMR of SAG in DMSO-d6.

Position	*δ*^13^C	*δ*^1^H (*J* in Hz)
2	164.3	
3	103.1	6.87 (1H, s)
4	182.0	
5	161.3	
6	99.6	6.46 (1H, d,2)
7	162.6	
8	94.7	6.83 (1H, d,2)
9	156.9	
10	105.4	
1′	121.0	
2′	128.5	7.96 (2H, d,8.8)
3′	115.9	6.95 (2 H, d,8.8)
4′	161.1	
5′	115.9	6.95 (2 H, d,8.8)
6′	128.5	7.96 (2H, d,8.8)
1″	99.6	5.12 (1H, d,7.2)
2″	73.0	3.30–3.45 (2H, m)
3″	76.2	3.30–3.45 (2H, m)
4″	69.8	3.15–3.30 (1H, m)
5″	74.0	3.70–3.85 (1H, m)
6″	63.6	4.43 (1H, d,10.4) 4.04–4.15 (1H, m)
1‴	172.0	
2‴	28.6	2.50–2.65 (2H, m)
3‴	28.6	2.35–2.0 (2H, m)
4‴	173.2	

**Table 2 molecules-28-00533-t002:** α-glucosidase inhibitory activity and docking analysis of active compounds.

Compound	Binding Energy (kcal/mol)
Apigenin	−9.0
AG	−8.9
SAG	−9.4
Acarbose	−8.0

**Table 3 molecules-28-00533-t003:** α-Glucosidase inhibitory activity analysis of active compounds.

Compounds	Inhibition, %			IC_50_, mM
	0.1 mM	0.5 mM	1 mM	
Apigenin	11.11 ± 0.70	58.33 ± 0.95	86.11 ± 0.07	0.525
AG	10.59 ± 0.08	56.65 ± 0.51	84.11 ± 0.10	0.532
SAG	16.67 ± 0.55	55.56 ± 0.35	88.89 ± 0.90	0.485
Acarbose	10.56 ± 0.11	41.67 ± 0.26	89.96 ± 0.24	0.554

## Data Availability

The data presented in this study are available on request from the corresponding author. The strains are deposited in the Jiangsu Collaborative Innovation Center of Chinese Medicinal Resources Industrialization. The NMR data obtained in this study are summarized in Table 1, and the MS and spectrum data are illustrated in the Appendix A.

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
