# Peer review of "Efficient Synthesis and In Vitro Hypoglycemic Activity of Rare Apigenin Glycosylation Derivatives"

_molecules, 2023, doi:10.3390/molecules28020533_

Round 1
Reviewer 1 Report
The manuscript describes novel and interesting method to obtain apigenin glycoside derivatives with apigenin aglycone as a substrate. Glycosylation resulted in the formation of AG and SAG which are more soluble in water than apigenin, which is also expected to increase bioavailability. Overall, the experiments are interesting but the manuscript could be improved with some clarifications and there are some suggested changes that might allow readers better follow what was done.
1. The authors do not describe by which method or test they determined cell viability. They only write "The HepG2 cells were co-cultured for 24 hours before adding 10 μL CCK-8 solution for 2 hours". What is CCK-8 solution?
2. In the results section, the authors describe subsection 3.4 Cell Proliferation Assay of Apigenin and Derivatives. The authors should decide whether they determined Hep-G2 cell viability or Hep-G2 cell proliferation using the CCK-8 solution.
3. Methodology 2.7 Assay of Cell Viability describes the addition of 5, 10, or 20 μM of apigenin, AG, or SAG to cells, meanwhile the results show concentrations of 5, 10, and 20 μmol/L. Please standardize the units.
4. Why Figure 2 is placed under the description of the results of the Cell Proliferation Assay of Apigenin and Derivatives?
5. In Table 3, the authors provide IC50 values for Apigenin, Apigenin 7-glucoside, SAG and Acarbose. How did the authors determine the reported IC50 values?
Author Response
Question 1. The authors do not describe by which method or test they determined cell viability. They only write "The HepG2 cells were co-cultured for 24 hours before adding 10 μL CCK-8 solution for 2 hours". What is CCK-8 solution?
Answer 1. We really appreciate that you made a very careful revision and valuable comments on our manuscript. In the revised manuscript, “The assay of cell viability was conducted according to the Cell Counting Kit-8 (CCK8) method as previously described using human hepatoma HepG2 cell lines.” was added to describe the method that we determined cell viability (Please see lines 156-157, marked with yellow).
Question 2. In the results section, the authors describe subsection 3.4 Cell Proliferation Assay of Apigenin and Derivatives. The authors should decide whether they determined Hep-G2 cell viability or Hep-G2 cell proliferation using the CCK-8 solution.
Answer 2. Thank you very much for your advice. We determined Hep-G2 cell viability using the CCK-8 solution, so we have revised “3.4 Cell Proliferation Assay of Apigenin and Derivatives” to “Cell viability Assay of Apigenin and Derivatives” and revised the figure note of Fig.3 according to your advice. ( Please see line 254,marked with yellow; figure 3)
Question 3. Methodology 2.7 Assay of Cell Viability describes the addition of 5, 10, or 20 μM of apigenin, AG, or SAG to cells, meanwhile the results show concentrations of 5, 10, and 20 μmol/L. Please standardize the units.
Answer 3. Thank you very much for your wonderful comments. We have standardized the units according to your advice. (Please see lines 260, 261, 279, 280, 282, 284, marked with yellow; figure 3-5)
Question 4. Why Figure 2 is placed under the description of the results of the Cell Proliferation Assay of Apigenin and Derivatives?
Answer 4. Thank you very much for your valuable advice. We apologized that we did not check the content of each part of the image carefully. We have adjusted figure 2 to the results of Molecular Docking. (Please see lines 233-237, Figure 2)
Question 5. In Table 3, the authors provide IC50 values for Apigenin, Apigenin 7-glucoside, SAG and Acarbose. How did the authors determine the reported IC50 values?
Answer 5. Thank you very much for your wonderful comments. Firstly, we examined the anti-α-glucosidase activity of three different concentrations of apigenin, AG and SAG. Based with the initial screening with activity, five concentrations at 0.1-1 mM of apigenin, AG and SAG were determined and the IC50 values were calculated as log-inhibition of concentrations using GraphPad Prism 8.01 software for graphing.
Reviewer 2 Report
Structures of the graphical abstract are not readable.
Multiple instances of full stop coming before citations in brackets at the end of sentences; should be after (lines 36, 38, 40, 51, 55, 58, 60, 63, 65, 68, 72, 139, 148, 161, 176, 194, 198, 216, 217, 221,264, 288, 289, 291, 300, 302, 304,312,
Names of compounds should be written correctly in carbohydrate nomenclature with -O- in cursive letters and D in small caps. e.g. line 13 apigenin-7-O-β-(6"-O)-D-glucoside; (lines 185, 254, 257)
Remove comma, line 74 before citation brackets
Multiple instances of comma coming before citations in brackets (lines 48, 77,217, 312)
Line 95: and other high-quality analytical grade reagents and solvents were from commercial sources. (which solvents and reagents from which sources? Not clear for me)
Comma missing between COVID-19 and brackets line 45
The study built with respect to a new biosynthetical way to AG and SAG with molecular docking and good yields.
Best regards,
Author Response
Response to REVIEWER 2
Question 1. Structures of the graphical abstract are not readable.
Answer 1. Thank you very much for your advice. We have corrected all the errors of structures in the graphical abstract and adjusted the resolution of the images. (Please see graphic abstract)
Question 2. Multiple instances of full stop coming before citations in brackets at the end of sentences; should be after (lines 36, 38, 40, 51, 55, 58, 60, 63, 65, 68, 72, 139, 148, 161, 176, 194, 198, 216, 217, 221,264, 288, 289, 291, 300, 302, 304,312)
Answer 2. Thank you very much for your advice. We have corrected all the errors of format in the revised manuscript carefully.
Question 3. Names of compounds should be written correctly in carbohydrate nomenclature with -O- in cursive letters and D in small caps. e.g. line 13 apigenin-7-O-β-(6"-O)-D-glucoside; (lines 185, 254, 257).
Answer 3. Thank you very much for your advice. We have corrected the format of “apigenin-7-O-β-(6"-O)-D-glucoside” to “apigenin-7-O-β-(6"-O)-D-glucoside” and revised “apigenin-7 -glucoside” to “AG”. (Please see lines 186,200-201,228, 235-237, 264-265, marked with yellow).
Question 4. Remove comma, line 74 before citation brackets.
Multiple instances of comma coming before citations in brackets (lines 48, 77,217, 312)
Answer 4. Thank you very much for your advice. We have adjusted the format of citations in brackets in the revised manuscript carefully.
Question 5. Line 95: and other high-quality analytical grade reagents and solvents were from commercial sources. (which solvents and reagents from which sources? Not clear for me)
Answer 5. Thank you very much for your advice. In the revised manuscript, we have added the sources of high-quality analytical grade reagents and solvents ,such as “Peptone,yeast extract, NaCl, Na2HPO4, KH2PO4 and sucrose in microbiological media were from Sinopharm Group Chemical Reagent Co. (Beijing, China).” ,“and methanol, formic acid in analytical grade were from Sinopharm Group Chemical Reagent Co. (Beijing, China).”,“Dulbecco’s modified Eagle medium (DMEM), fetal bovine serum (FBS), penicillin, and streptomycin were purchased from Gibco (Invitrogen, Carlsbad, CA, USA). The assay of cell viability was estimated with a Cell Counting Kit-8 (APEXBIO, USA).”.(Please see lines 93-95,96-97,100-104, marked with yellow)
Question 6. Comma missing between COVID-19 and brackets line 45.
Answer 6. Thank you very much for your advice. We have added the comma between COVID-19 and brackets. (Please see line 45, marked with yellow)

Reviewer 3 Report
The article "Efficient Synthesis and In Vitro Hypoglycemic Activity of Rare Apigenin Glycosylation Derivatives" by Zhang, Duan an coworkers present the preparation of rare apigenin glycosyl and succinyl derivatives by bacteria Bacillus licheniformis WNJ02 and the in vitro evaluation of their anti-a-glucosidase activity. The results obtained were in accordance with the docking studies performed.
The initial hypothesis is clearly stablished and it has been fulfilled with the studies performed. However, several issues need to be clarified and/or modified in order to make it acceptable for publication.
Major points:
- A figure with the structures of the flavonoids object of study should be included in the introduction.
- The term SAG should be described in the introduction as it has been done with AG.
- The authors refer to AG and SAG as "rare glycosylation derivatives." This should be justified.
- The authors have previously reported the preparation of SAG by using B. amyloliquefaciens FJ18 (ref. 20). The novelty of the methodology reported in the article by using WNJ02 should be justified.
- The chemical structure for apigenin is wrong. In consequence, the structures for AG and SAG have to be corrected as well and revised throughout the article. The caption of Figure 1 is unclear. The numbering of the compound structure used for the NMR characterization should be included in Scheme 1.
- The characterization of SAG needs revision. The molecular formula and notation of the molecular ion are incorrect as according with the supporting information, the authors have provided the mass spectra in positive mode. Additionally, in Table 1, the total number of protons should be revised.
- The term "anti-a-glucosidase inhibitor" in lines 22, 194 and 200 is misleading.
- The colour code used for the interactions in the docking studies should be included in the caption of Figure 2. The electrostatic interactions for AG are missing in the text and the formation of hydrogen bonds with GLU374 in the case of AG is indicated twice (lines 206-209).
- The IC50 value for AG written in the text differs from the one indicated in the corresponding table, please unify. The differences between the IC50 values are small (0.485 mM vs. 0.525 and 0.532 mM), therefore the term "much lower" should be pondered.
- The authors indicate in lines 323-324 that their method is greener and much efficient compared with the existing systems. Please provide additional explanation.
Minor points:
- The article should be revised and typos corrected (for example in lines 18, 177, 214, 302 and 305).
- In some references, DOI number is missing and the journal name is not properly abbreviated.
Author Response
Response to REVIEWER 3
Question 1. A figure with the structures of the flavonoids object of study should be included in the introduction.
Answer 1. Thank you very much for your advice. The figure 1 has showed the structures of the flavonoids of study. In this study, we bio-transformed apigenin into AG and SAG with exclusive strain, which is shown in figure 1. In the revised manuscript, “as shown in figure 1” has been added. (Please see line 80, marked with yellow)
Question 2. The term SAG should be described in the introduction as it has been done with AG.
Answer 2. Thank you very much for your advice. SAG is a novel compound synthesized by our team for the first time, so there are no further reports on this compound. In the revised manuscript, “SAG is a novel compound synthesized by our team for the first time, meanwhile the hypoglycemic activity of SAG has not been reported” has been added to describe the term SAG.(Please see lines 77-78,marked with yellow)
Question 3. The authors refer to AG and SAG as "rare glycosylation derivatives." This should be justified.
Answer 3. Thank you very much for your advice. Apigenin-7-O-glucoside is an active phenolic compound in nature plants and possesses remarkable therapeutic applications. However, the high price of AG and low abundance in plants limit its use, meanwhile it would hydrolyze in the purification process[1], so we referred to AG as rare glycosylation derivatives. SAG is a novel compound synthesized by our team for the first time, so we also considered SAG to be rare glycosylation derivatives. (Please see lines 46-49, 77-78, marked with yellow)
- Wang, Y.; Xu, Z.; Huang, Y.; Wen, X.; Wu, Y.; Zhao, Y.; Ni, Y. Extraction, Purification, and Hydrolysis Behavior of Apigenin-7-O-Glucoside from Chrysanthemum Morifolium Tea. Molecules 2018, 23, E2933, doi:10.3390/molecules23112933.
Question 4. The authors have previously reported the preparation of SAG by using B. amyloliquefaciens FJ18 (ref. 20). The novelty of the methodology reported in the article by using WNJ02 should be justified.
Answer 4. Thank you very much for your advice. We will introduce the innovation of this research from the following three points. First of all, the innovation of strains, FJ18 is Bacillus amyloliquefaciens in our previous research, while WNJ02 is Bacillus licheniformis, whose enzyme stability is stronger than that of Bacillus amyloliquefaciens and its application is more extensive[2]. Secondly, the synthesis and neuroprotection of novel apigenin glycoside derivative produced by FJ18 were mainly introduced in previous study. In this study, WNJ02 was used to achieve higher conversion efficiency, with a conversion rate of 94.2%[3]. What’s more, WNJ02 can efficiently synthesize AG and SAG at the same time. (Please see lines 185-191, marked with yellow; Figure 1)
- Ahmad, A.; Mishra, R. Different Unfolding Pathways of Homologous Alpha Amylases from Bacillus Licheniformis (BLA) and Bacillus Amyloliquefaciens (BAA) in GdmCl and Urea. Int J Biol Macromol 2020, 159, 667–674, doi:10.1016/j.ijbiomac.2020.05.139.
- Zhang, S.; Xu, S.; Duan, H.; Zhu, Z.; Duan, J. A Novel, Highly-Water-Soluble Apigenin Derivative Provides Neuroprotection Following Ischemia in Male Rats by Regulating the ERK/Nrf2/HO-1 Pathway. European Journal of Pharmacology 2019, 855, doi:10.1016/j.ejphar.2019.03.024.
Question 5. The chemical structure for apigenin is wrong. In consequence, the structures for AG and SAG have to be corrected as well and revised throughout the article. The caption of Figure 1 is unclear. The numbering of the compound structure used for the NMR characterization should be included in Scheme 1.
Answer 5. Thank you very much for your advice. We have corrected all the errors in the revised manuscript carefully and revised the caption of figure1. (Please see lines 188-191, marked with yellow; Figure 1a and Figure 1b)
Figure 1. a: Synthesis of apigenin-7-O-β-(6″-O)-D-glucoside (AG) and apigenin-7-O-β- (6″-O-succinyl)-D-glucoside (SAG), b: The structure of SAG with numbers.
Question 6. The characterization of SAG needs revision. The molecular formula and notation of the molecular ion are incorrect as according with the supporting information, the authors have provided the mass spectra in positive mode. Additionally, in Table 1, the total number of protons should be revised.
Answer 6. Thank you very much for your advice. We have corrected the molecular formula, notation of the molecular ion and the total number in the revised manuscript carefully. ( Please see lines 188-191, marked with yellow)
Question 7. The term "anti-α-glucosidase inhibitor" in lines 22, 194 and 200 is misleading.
Answer 7. Thank you very much for your advice. We have modified all the term “anti-a-glucosidase inhibitor” to “α-glucosidase inhibitor” in the revised manuscript. (Please see lines 22, 206, 212, marked with yellow)
Question 8. The colour code used for the interactions in the docking studies should be included in the caption of Figure 2. The electrostatic interactions for AG are missing in the text and the formation of hydrogen bonds with GLU374 in the case of AG is indicated twice (lines 206-209).
Answer 8. Thank you very much for your advice. We have added the colour code used for the interactions in the docking studies in the caption of Figure 2. For a more rigorous presentation, the results of docking of AG have been adjusted “AG interacts with 7D9B, mainly through the formation of hydrogen bonds as well as hydrophobic forces, formed hydrogen bonds with residues GLU349, ASP440 and ARG488; and hydrophobic interactions with VAL346, TRP376, HIS439 and HIS214.” and Figure 2b has also been corrected because of the structure of AG. (Please see lines 217-220, marked with yellow; Figure 2)
Question 9. The IC50 value for AG written in the text differs from the one indicated in the corresponding table, please unify. The differences between the IC50 values are small (0.485 mM vs. 0.525 and 0.532 mM), therefore the term "much lower" should be pondered.
Answer 9. Thank you very much for your advice. We have unified all the errors and delete the “much” for a more rigorous comparison in the revised manuscript. (Please see line 246 marked with yellow)
Question 10. The authors indicate in lines 323-324 that their method is greener and much efficient compared with the existing systems. Please provide additional explanation.
Answer 10. Thank you very much for your advice. In this study, WNJ02 in our method was used to achieve higher conversion efficiency, with a conversion rate of 94.2 %, and it can efficiently synthesize AG and SAG at the same time. What’s more, compared with chemical or enzymatic engineering methods for succinylation, synthesizing succinyl groups also suffers from cumbersome experimental procedures and specificity of donor substrates, which highlights the advantages of exclusive strain biotransformation. Meanwhile, there is no hazardous and toxic chemical reagents in the exclusive strain biotransformation and the conversion of apigenin was 94.2% in 24 h, so we think our method in this study is greener and much efficient compared with the existing systems. (Please see lines 108-113, marked with yellow)
Question 11. The article should be revised and typos corrected (for example in lines 18, 177, 214, 302 and 305).
Answer 11. Thank you very much for your advice. We have corrected all the errors in the revised manuscript carefully and will improve the ability of language and academical writing. (Please see lines 12, 18, 183, 310, marked with yellow)
Question 12. In some references, DOI number is missing and the journal name is not properly abbreviated.
Answer 12. Thank you very much for your advice. We have corrected the DOI number and the journal name in the revised manuscript carefully. The correction is as follows:
- Patel, D.; Shukla, S.; Gupta, S. Apigenin and Cancer Chemoprevention: Progress, Potential and Promise (Review). Int J Oncol 2007, 30, 233–245, doi: 10.3892/ijo.30.1.233.

Round 2
Reviewer 3 Report
The authors have adressed the points raised in the previous revision.